# Effect of Printing Direction on the Accuracy of 3D-Printed Dentures Using Stereolithography Technology

**DOI:** 10.3390/ma13153405

**Published:** 2020-08-02

**Authors:** Tamaki Hada, Manabu Kanazawa, Maiko Iwaki, Toshio Arakida, Yumika Soeda, Awutsadaporn Katheng, Ryosuke Otake, Shunsuke Minakuchi

**Affiliations:** 1Gerodontology and Oral Rehabilitation, Graduate School of Medical and Dental Sciences, Tokyo Medical and Dental University, Yushima, Bunkyo, Tokyo 113-8549, Japan; t.hada.gerd@tmd.ac.jp (T.H.); aragerd@tmd.ac.jp (T.A.); y.soeda.gerd@tmd.ac.jp (Y.S.); katheng.gerd@tmd.ac.jp (A.K.); r.otake.gerd@tmd.ac.jp (R.O.); s.minakuchi.gerd@tmd.ac.jp (S.M.); 2General Dentistry, Graduate School of Medical and Dental Sciences, Tokyo Medical and Dental University, Yushima, Bunkyo, Tokyo 113-8549, Japan; m.iwaki.gerd@tmd.ac.jp

**Keywords:** accuracy verification, complete denture, precision, printing, stereolithography, three-dimensional trueness

## Abstract

This study evaluated the effects of the differences in the printing directions of stereolithography (SLA) three-dimensional (3D)-printed dentures on accuracy (trueness and precision). The maxillary denture was designed using computer-aided design (CAD) software with an STL file (master data) as the output. Three different printing directions (0°, 45°, and 90°) were used. Photopolymer resin was 3D-printed (n = 6/group). After scanning all dentures, the scanning data were saved/output as STL files (experimental data). For trueness, the experimental data were superimposed on the master data sets. For precision, the experimental data were selected from six dentures with three different printing directions and superimposed. The root mean square error (RMSE) and color map data were obtained using a deviation analysis. The averages of the RMSE values of trueness and precision at 0°, 45°, and 90° were statistically compared. The RMSE of trueness and precision were lowest at 45°, followed by 90°; the highest occurred at 0°. The RMSE of trueness and precision were significantly different among all printing directions (*p* < 0.05). The highest trueness and precision and the most favorable surface adaptation occurred when the printing direction was 45°; therefore, this may be the most effective direction for manufacturing SLA 3D-printed dentures.

## 1. Introduction

With the recent and rapid advances in digital technology, new digital tools called computer-aided design (CAD) and computer-aided manufacturing (CAM) have become popular in the field of dentistry [1,2]. The designing and manufacturing of prosthetic devices using such digital tools have helped to reduce the burden on dentists and dental technicians [3]. There are two main processing methods for CAD-CAM systems: subtractive manufacturing (SM) and additive manufacturing (AM).

SM has been used for the production of dental prostheses such as crown bridges and implants, and its effectiveness has been confirmed. Additionally, SM has been actively applied to removable dentures [4,5,6,7,8]. The most common method involves designing the complete denture using CAD software, milling the denture base using a denture base resin disc, and bonding the existing artificial teeth [5,6]. This method of milling a ready-made disk can be processed with high accuracy because there is no polymerization shrinkage of the material itself, although it depends on the size and number of milling burs [9,10]. However, since only one denture base can be machined per denture base resin disc, the disc has many surplus parts after milling. Furthermore, milling burs are prone to becoming extremely worn out because the time required for milling one denture base can be long, which is not economical.

AM is a method of manufacturing a dental prosthesis by laminating and molding photopolymer resin and metal powder. Recently, a method of printing artificial teeth and a denture base separately using a three-dimensional (3D) printer and bonding the two with a photopolymer resin has been practically used [11,12]. The cost of 3D printers is low, and the printing time is short and economical; therefore, we have greater opportunities to use 3D printing in dental laboratories and dental clinics [13]. Typical examples of modeling methods for 3D printers are selective laser melting (SLM) for powder bed fusion [14,15], fused deposition modeling (FDM) for material jetting [16] and material extrusion [17], and stereolithography (SLA) and digital light processing (DLP) for vat photopolymerization. These methods are classified into seven types according to the ISO standard (ISO17296-2:2015 AM Part 2: Overview of process categories and feedstock). SLA is a method involving laser beam raster scanning of the surface of a liquid tank filled with a methacrylate-based photocurable resin to create a model [18]. The advantage of this method is that it is possible to produce transparent objects and create large, high-resolution models. Furthermore, by setting detailed parameters such as the print pitch, it is possible to form complex shapes with undercuts that are difficult to shear, and further applications in the dental field are expected [19].

Benefits to using 3D printing to create complete dentures are the quicker manufacturing time and less expensive cost compared to those of conventional methods [20]. In general, the number of models that can be planted on a platform depends on the shape and printing direction; vertical printing can produce more models than horizontal printing. There have been many studies of SLA 3D printers because their accuracy in creating printed objects differs on the basis of the printing direction, parameter settings, and the type of material used. Unkovskiy et al. [21] examined the precision of bar-shaped test specimens based on the printing direction used. Tahayeri et al. [22] examined the precision when changing the printing direction and parameters such as light intensity, exposure time, and slice thickness during the manufacturing of bar-shaped test specimens using various 3D printer materials. These studies demonstrated that several factors, such as the parameters and printing directions of the SLA 3D printer, affect precision. However, most specimens have simple shapes such as a bar [21,22], crown [23,24], and prism [25]. No studies have evaluated the effects of the differences in the printing directions (0°, 45°, and 90°) (Figure 1) on the accuracy (trueness and precision) of clinical dentures. Therefore, this study aimed to evaluate the effects of differences in the printing directions (0°, 45°, and 90°) on the accuracy (trueness and precision) of SLA 3D printing when creating dentures. The null hypothesis was that the three different printing directions would not show any differences in accuracy (trueness and precision).

## 2. Materials and Methods

The flow chart of the SLA 3D printing accuracy verification protocol for dentures is shown in Figure 2. The surface of the maxillary edentulous model (G2-402U; Nissin Dental Products, Kyoto, Japan) and the artificial teeth (Velasia SA Antieria A3 ST5; Shofu, Kyoto, Japan) were coated with scanning powder (high-resolution scanning spray; 3M, St. Paul, MN, USA) and scanned with a 3D optical scanner (NeWay; Open Technologies, Rezzato, Italy) to obtain surface image data. A maxillary denture was designed by CAD software (Geomagic Freeform; 3D Systems, Rock Hill, SC, USA). The artificial teeth data were deleted from the denture data using Boolean logic operations, and a socket part for inserting the artificial teeth into the denture base was prepared. Then, the denture base was saved as a standard tessellation language (STL) file (master data) and sent to the 3D printing software (PreForm Software; Formlabs, Somerville, MA, USA). Three groups of dentures were assigned according to three different printing directions (Figure 3): 0°, print layer vertical to the z-axis direction (n = 6); 45°, print layer 45° to the z-axis direction (n = 6); and 90°, print layer horizontal to the z-axis direction (n = 6). A commercially available methacrylate-based photopolymer resin (Clear; Formlabs) was printed using a desktop SLA 3D printer (Form 2; Formlabs). Eighteen dentures with a thickness layer of 100 µm (z resolution) were fabricated (n = 6 per group). Table 1 and Table 2 show the composition and processing parameters of this photopolymer resin. The dentures were washed with isopropyl alcohol (99.9 %) for 15 min (Form Wash; Formlabs) to remove excess resin. Once they were completely dry, polymerization was performed at 60 °C for 10 min in a 350- to 500-nm wavelength light-emitting diode oven (Form Cure; Formlabs) according to the manufacturer’s instructions. Up to 24 h before the start of measurements, the denture base was stored in a lightproof container at a constant temperature of 23 °C.

### 2.1. Accuracy Verification

In this study, materials and measuring equipment were calibrated according to the manufacturer’s recommendations to precisely analyze the accuracy of the denture and eliminate bias in measurement methods. After 3D printing, the supporting materials were removed, and a 3.5-magnification optical loupe was used to visually inspect the denture surface for defects. The mucosal surface of the denture base was coated with scanning powder (high-resolution scanning spray; 3M) with an average particle size of 3.0 µm. All dentures were scanned with a 3D optical scanner (NeWay; Open Technologies). Scanning data were saved/output as STL files (experimental data). The International Standards Organization defined both trueness (closeness of measured values to the true value) and precision (closeness of measured values during repeated measurements) as accuracy (ISO 5725-1) [26]. During accuracy research of 3D printing, it was observed that the trueness value increased when the printed object and the CAD-designed object were dimensionally close, and that the precision value increased when the printed objects were dimensionally close. During the trueness test, experimental data were superimposed on master data (six patterns per direction). During the precision test, two sets of experimental data were selected from six dentures created using three different printing directions and superimposed on each other (15 combinations per direction). Superposition was performed automatically using the iterative tangent point algorithm after manually aligning the data along the flange of the internal surface to create the same coordinate system. A deviation analysis was performed using 3D analysis software (CATIA V5; Dassault Systèmes, Vélizy-Villacoublay, France). The distance between each set of data obtained by the deviation analysis was calculated and output as a text file. The root mean square error (RMSE) value (mm) was used to quantify the trueness and precision and was calculated using the following formula:(1)$$\mathrm{RMSE}=\frac{1}{\sqrt{n}}\cdot \sqrt{\sum_{i=1}^{n}{\left(x_{1,i}-x_{2,i}\right)}^{2}}$$
where *x_1_*,*_i_* refers to the measurement point *_i_* in the master data, *x_2_*,*_i_* refers to the measurement point of *_i_* in the experimental data, and *n* refers to the total number of points. The average of the RMSE values calculated for each denture was considered the representative value. Subsequently, a color map was created for qualitative expression. The nominal deviation was set to ±100 µm, and the maximum critical deviation was set to ±300 µm. The color map shows the average deviation between experimental data and master data for trueness and the average deviation among experimental data for precision as follows: positive deviation, yellow to red (area where the experimental data were larger than the master data and exceed the limit of the allowable range [100 µm]); negative deviation, light blue to blue (area where the experimental data were smaller than the master data and exceed the lower limit of the allowable range [−100 µm]); and acceptable deviation, light green to green.

### 2.2. Statistical Analyses

Statistical analyses were performed using statistical analysis software (IBM SPSS statistics 22.0; IBM Corp., Armonk, NY, USA). The Shapiro-Wilk test found normality in the data distribution, and the homogeneity of variance was satisfied according to Levene’s test. Therefore, for trueness and precision, the averages of the RMSE values at 0°, 45°, and 90° were statistically compared using one-way analyses of variance (ANOVA) and Turkey’s test for multiple comparisons with a significance level of α = 0.05.

## 3. Results

The RMSE values of trueness were the lowest for dentures printed at 45° (0.086 ± 0.004 mm; *p* = 0.001), followed by those for dentures printed at 90° (0.109 ± 0.005 mm; *p* = 0.001) and at 0° (0.129 ± 0.006 mm; *p* = 0.001) (Table 3). The RMSE values for trueness were the highest at 0°. There was a significant difference in the RMSE values for trueness among all printing directions (*p* < 0.05). The color map shows that deviations existed in both positive and negative directions (Figure 4). The palate showed a light green color under all conditions, and the part equivalent to the ridge showed a green color (acceptable deviation). At 0°, yellow and red colors (positive deviation) were observed from near the incisive papilla to the left and right, and a light blue-to-blue color (negative deviation) appeared prominently at the central palate and denture border. At 45°, a yellow color (positive deviation) was observed near the linguogingival residual, and a light blue color (negative deviation) was observed near the palate folds and labial denture border. At 90°, a yellow color (positive deviation) was noticeable at the part equivalent to the incisive papilla and pterygomandibular fold, and a light blue-to-blue color (negative deviation) appeared near the palatine foveola. The RMSE values of precision were lowest for dentures printed at 45° (0.050 ± 0.003 mm; *p* = 0.001), followed by those printed at 90° (0.069 ± 0.002 mm; *p* = 0.001) and at 0° (0.072 ± 0.004 mm; *p* = 0.001) (Table 4). The RMSE values of trueness were the highest at 0°, and there was a significant difference between the RMSE values for precision using all printing directions (*p* < 0.05). The color map showed colors from light green-to-green (acceptable deviation) for most of the printing directions (Figure 5). Almost uniform color appeared at 45°. While observing the part where the deviation appeared in the color map, a staircase effect [27,28,29,30,31,32,33,34,35] of the print layer that could be confirmed with the naked eye occurred frequently (Figure 6). Positive deviations had a concave staircase effect, and negative deviations had a convex staircase effect (Figure 7).

Experimental data were superimposed on the master data, and the distance between each data point was obtained by the deviation analysis of six patterns (n = 6) for each direction. The root mean square error (RMSE) in mm was subsequently calculated.

Two sets of experimental data were selected from six dentures in the three different printing directions and superimposed on each other. The distance between each data point was obtained by a deviation analysis of 15 combinations for each direction (n = 15). The root mean square error (RMSE) in mm was subsequently calculated.

## 4. Discussion

During this in vitro study, we investigated the differences in the three printing directions when creating dentures with a 3D printer and found that the accuracy (trueness and precision) was significantly different with each direction. Therefore, the null hypothesis that three different printing directions used for SLA 3D printing dentures do not affect accuracy (trueness and precision) was rejected.

In terms of trueness, the mean deviation of the RMSE values in this study was 0.086–0.129 mm; however, the mean deviation of the RMSE values in a previous in vitro study [32] that evaluated the trueness of 3D printing full-arch models was 0.085 mm. The trueness of our study was similar to that reported in that previous study [32].

In this study, the RMSE values of trueness were largest when printing was performed at 0° in the z-axis direction (the print layer vertical to the z-axis direction). In other words, the fit of the denture printed at 0° was the least clinically favorable among the three printing directions. According to the color map of the deviation in the trueness of the mucosal surface of the denture (Figure 4), at 0°, a positive deviation was noticeable from near the incisive papilla to the left and right, and a negative deviation was prominent in the central palate and denture border. Positive deviations indicate a gap between the experimental data and the master data, suggesting that poor matching between the denture and mucosa may lower the stability and maintenance force of the denture. Negative deviations indicate that the master data are smaller than the experimental data, suggesting that denture adjustment may be necessary because of the strong contact between the denture and mucosa. According to a previous study [27], the surface accuracy of a printed object depends on the surface roughness. Furthermore, the staircase effect was caused by the thickness of the printed layer and the gradient of the surface of the object [27,29,30]. The maximum deviation between the surface of the printed layer and the CAD model surface caused by the staircase effect was expressed as the cusp height (CH) [29,30,31]. The CH depends on the printing direction (d), the angle (θ) formed by the normal CAD model surface, and the thickness (t) of the printed layer [CH = t cos (θ)]. A thick print layer and/or large cos (θ) value generated large CH and reduced the surface accuracy [33,34]. Therefore, Dolenc et al. [35] concluded that it was necessary to reduce the staircase effect to improve the surface quality of the object. In this study, when a denture printed at 0° with a constant print layer thickness (100 µm), a large cos (θ) value was generated at the part equivalent to the ridge crest, palate, and denture border. As a result, it was suggested that a remarkable staircase effect appeared in these three parts, and that positive and negative deviations were observed.

In our study, the RMSE values of trueness were minimal when printed at 45° in the z-axis direction (Table 3). In other words, dentures printed at 45° showed the best clinical fit among the three printing directions. A previous study [31] reported that printing an object at an angle reduced the surface area of each printed layer, thereby minimizing the ability to remove the object from the resin tank and greatly increasing the success rate of printing; the conclusion of that study [31] was consistent with that of our study, which indicated that printing at 45° showed the best accuracy. The color map of the deviation (Figure 4) showed a positive deviation near the linguogingival residual, suggesting the possibility of lowering the retention of the denture. No staircase effect was found when observing the part where the deviation appeared. Therefore, when observing the polished surface side, many support structures were denser than those created using the other two printing directions (0° and 90°). This was consistent with a previous study [24] that found positive deviations in the area with more support structures and the opposite side. When the part had many support structures, the part shrunk to the side of the support structure because it was damaged by more ultraviolet exposure than the less dense areas of the support structure. Therefore, it was suggested that the positive deviation that appeared near the linguogingival residual affected the support structures densely planted on the opposite side of the mucosal surface. However, negative deviations appeared near the palate folds and the labial denture border. Observation of this part showed a slight convex staircase effect. When dentures were printed at 45°, a large cos (θ) occurred near the palate folds and the labial denture border. As a result, a notable staircase effect appeared in these two places, suggesting a negative deviation.

The RMSE values of the trueness of the dentures printed at 90° were smaller than the RMSE values when dentures were printed at 0° and larger than the RMSE values when dentures were printed at 45° (Table 3). In other words, the fit of dentures printed at 90° was better than that of those printed at 0° and worse than that of those printed at 45°. A previous study [30] found that if the printing direction exceeded 45°, then overhang areas (a shape that cannot be supported without a support material) occurred, and it was necessary to add support structures to the surface of the object, which could adversely affect the surface accuracy of the object. Furthermore, in this study, the support structures were added not only to the polished surface side of the dentures printed at 90° but also to a part of the mucosal surface side to avoid displaying the printing error (overhang area) of the 3D printing software (PreForm Software; Formlabs). This suggested that the RMSE values of the trueness of the denture printed at 90° were larger than the values of those printed at 45°. In the color map of the deviation that occurred when printing was performed at 90° (Figure 4), the positive deviations were prominent at the part equivalent to the incisive papilla and the pterygomandibular fold, and a negative deviation was prominent near the palatine foveola. Observation of this area revealed the concave staircase effects (positive deviations) in the area corresponding to the part equivalent to the incisive papilla. When dentures were printed at 90°, a large cos (θ) occurred at the part equivalent to the incisive papilla. As a result, staircase effects appeared, suggesting a significant positive deviation. However, there were no staircase effects near the pterygomandibular fold and the palatine foveola where the positive and negative deviations appeared. Observation of this area revealed many additional support structures. Therefore, positive and negative deviations were suggested to be technical errors due to removing extra support structures to avoid overhang areas.

Regarding precision, the average deviations of the RMSE values in this study were 0.050–0.072 mm; however, in a previous in vitro study [23] that evaluated the precision of the crown fabricated by the SLA 3D printer using RMSE values, the maximum RMSE value was 0.042 mm. The difference was approximately 0.03 mm, which is clinically acceptable. Therefore, the precision of the SLA 3D printer used this time was relatively good. The RMSE value for precision was the least when dentures were printed at 45° in the z-axis direction, and it was the highest when dentures were printed at 0° in the z-axis direction (print layer vertical to the load direction). This suggested a tendency similar to the RMSE values for trueness. A comparison of the differences between the RMSE values for precision and the RMSE values for trueness showed that the RMSE values for precision were lower than the RMSE values for trueness by approximately 55–58%. It was suggested that the reason why the RMSE values for precision became small was that the dentures were printed near the center with high accuracy, thus avoiding the edge [21] of the platform where the accuracy was poor when printing the dentures. The color map of the deviation in precision (Figure 5) showed a green color in most areas in each of the printing directions. This was because the trueness of the experiment data was superimposed on the master data using CAD software, and the precision of the experiment was due to the superimposition of experimental data and other experimental data combined. Therefore, it was suggested that the precision of the printed objects showed the same tendency.

A post hoc power analysis was performed using the analysis software (G*Power 3.1.9.4 software; Kiel University, Kiel, Germany). Because the significance level was α = 0.05 in this study, the power of 1-β was 0.8 or more; therefore, the sample size in this study was appropriate. To verify accuracy, the RMSE values were used to numerically evaluate the trueness and precision [36]. The RMSE value is widely used to quantitatively evaluate dimensional accuracy by calculating the phase difference between each point of the denture data of the three-dimensional space [37].

Reproducing the denture morphology designed by CAD with the highest accuracy is important for accurately fitting the denture in the patient’s mouth and enabling long-term use [9]. The findings suggested that investigating differences in printing directions was essential to determining the overall clinical performance of 3D printing dentures. In this in vitro study, we selected the SLA 3D printer, which is widely used in various fields, and a corresponding photopolymer resin. The denture base after polymerization was stored in a lightproof container to prevent deformation due to ultraviolet rays [21,23].

The results of this study indicated that printing at 45° was recommended to minimize the effects on the accuracy of 3D printing dentures using SLA technology. If the denture printing direction deviates from this range, the denture will still be usable; however, to improve the accuracy of the denture, the printing direction must be at least 45°–90°.

The main limitation of this study was its in vitro nature. In this study, we used one type of SLA 3D printer and photopolymer resin corresponding to the model. The parameters of this model were fixed and could not be freely adjusted. Therefore, the same tendency was not always observed when using other types of SLA 3D printers. Clear photopolymer resin was chosen for this study because it was an in vitro study. We also conducted material property tests of the same materials and designs [38]. Therefore, we believe it is important to verify the accuracy using standardized materials and designs. However, there are various types of photopolymer resins other than those used in this study, and some resins could be used in open mode in printers made by other companies. Therefore, it is necessary to investigate the method that can produce the most accurate denture with various types of 3D printers, photopolymer resins, or a combination of these. The creation of dentures with 3D printers is simple, quick, and inexpensive compared to other conventional methods; therefore, it is expected that 3D printing will become more popular in the future to reduce the burden on dentists and dental technicians. Further studies investigating the effects of different 3D printing directions on accuracy in vivo are required.

## 5. Conclusions

The accuracy (trueness and precision) of dentures printed using an SLA 3D printer depends on the denture printing direction. The 45° printing direction showed the highest accuracy compared to the 0° and 90° printing directions.

## Figures and Tables

**Figure 1 materials-13-03405-f001:**
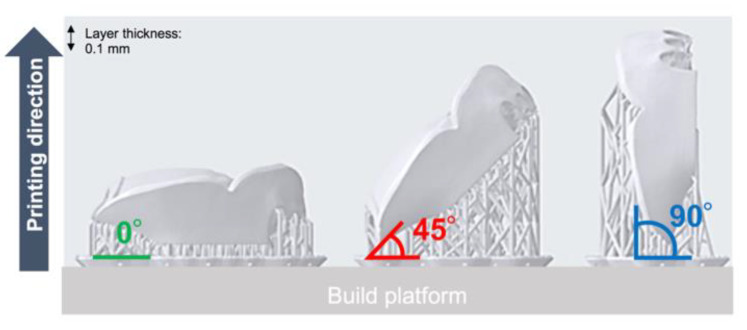
Schematic of three different print directions (0°, 45°, and 90°).

**Figure 2 materials-13-03405-f002:**
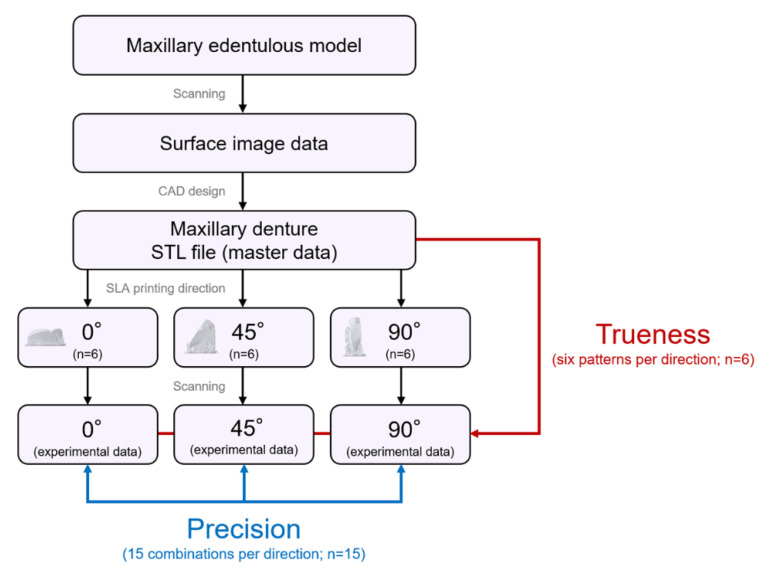
Flow chart of the stereolithography (SLA) 3D printing accuracy verification protocol for dentures.

**Figure 3 materials-13-03405-f003:**
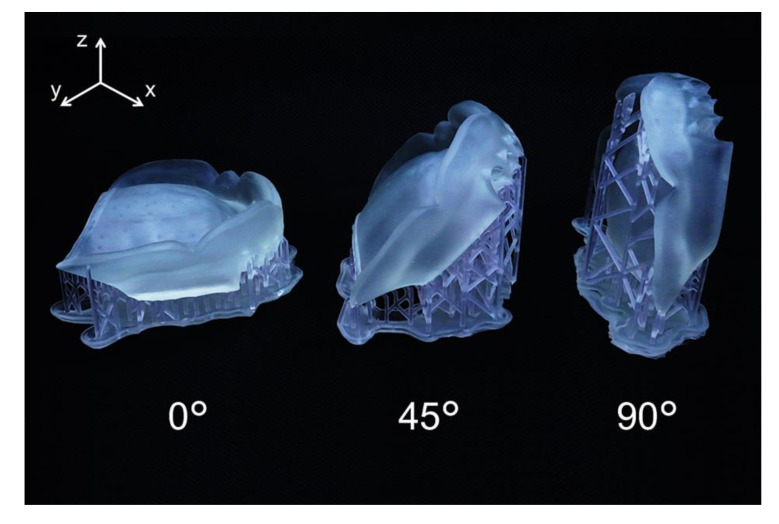
Dentures created using 3D printing with three different printing directions.

**Figure 4 materials-13-03405-f004:**
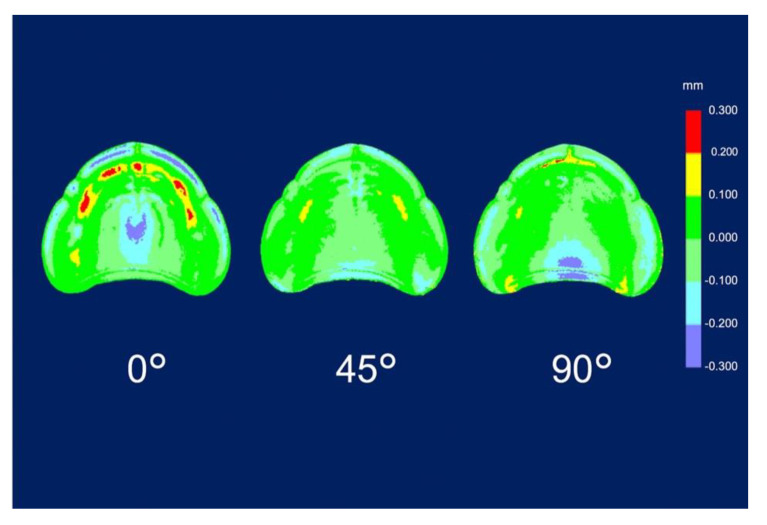
Deviation analysis of trueness. Color map deviation patterns of the denture mucosal surface with different printing directions. The trueness indicates the average deviation between the experimental data and the master data as follows: positive deviation, yellow to red; negative deviation, light blue to blue; and acceptable deviation, light green to green.

**Figure 5 materials-13-03405-f005:**
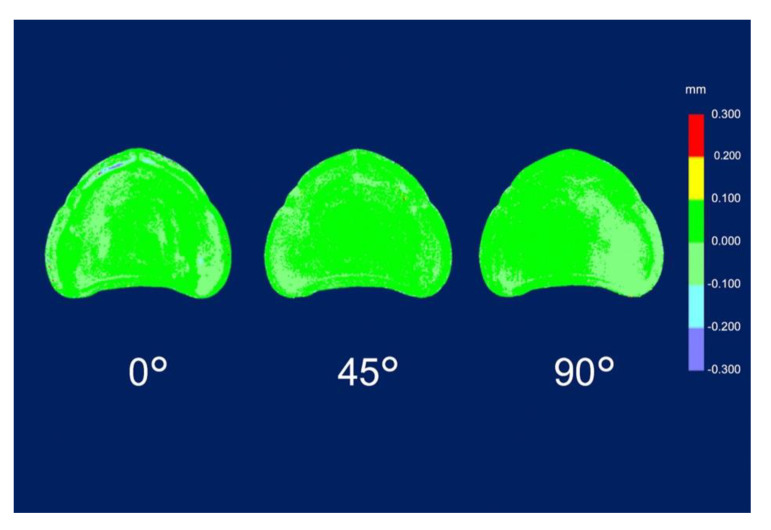
Deviation analysis of precision. Color map deviation patterns of the denture mucosal surface with different printing directions. The precision indicates the average deviation among the experimental data as follows: positive deviation, yellow to red; negative deviation, light blue to blue; and acceptable deviation, light green to green.

**Figure 6 materials-13-03405-f006:**
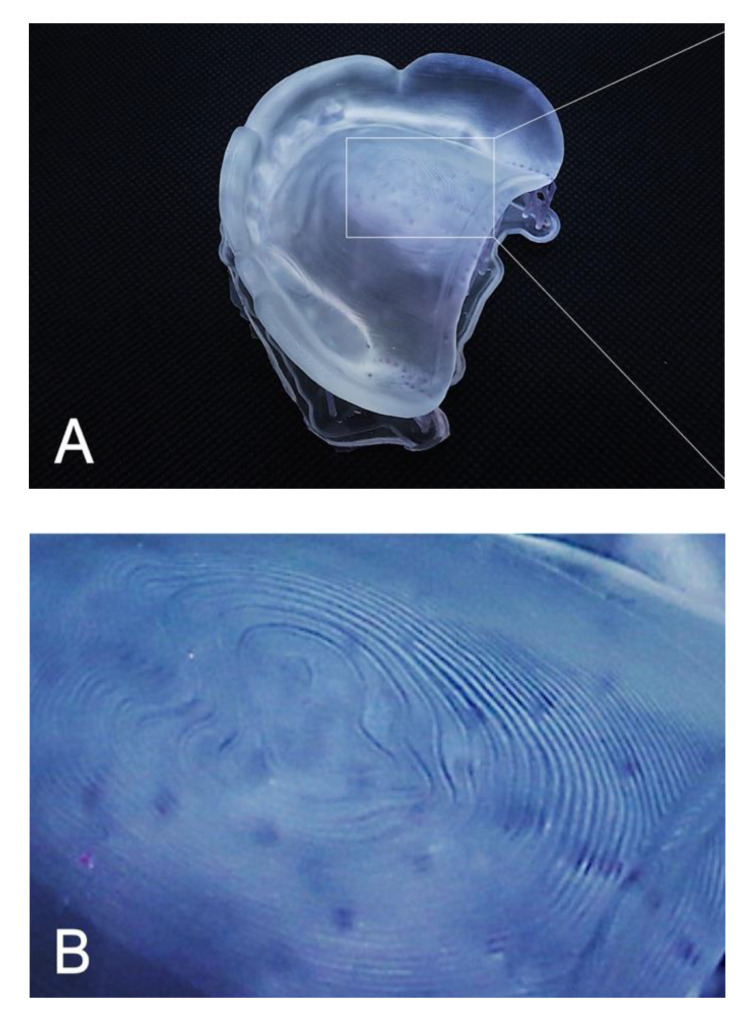
Staircase effects of the print layers on the surface of the denture mucosa. Many appeared when the printing direction was 0° and were clearly visible to the naked eye. (**A**) Total image. (**B**) Enlarged image of the staircase effects.

**Figure 7 materials-13-03405-f007:**
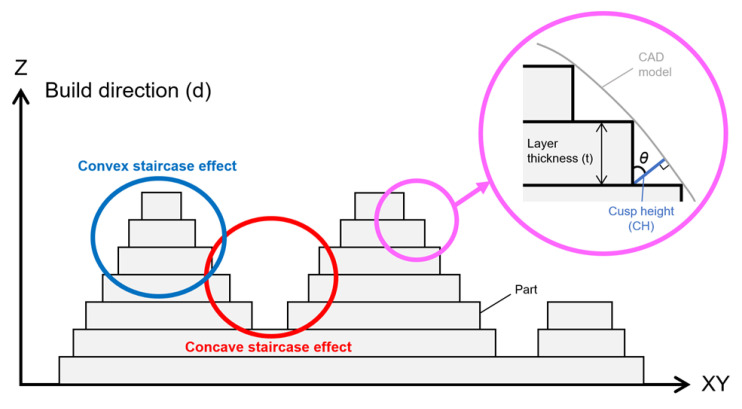
Concave staircase effects (positive deviation: red) and convex staircase effects (negative deviation: blue) in the trueness color map. The maximum deviation between the surface of the print layer and the CAD model surface caused by the staircase effect was called the cusp height (CH) [29,30,31]. The CH depends on the printing direction (d), the angle (θ) formed by the normal CAD model surface, and the thickness (t) of the printed layer. A thick print layer and/or a large cos (θ) value generated a large CH and reduced the surface accuracy.

**Table 1 materials-13-03405-t001:** Composition of transparent photopolymer resin.

Composition	Value (%)
Methacrylated oligomer	≥75–≤90
Methacrylated monomer	≥25–≤50
Diphenyl (2,4,6-trimethylbenzoyl) phosphine oxide	<1

**Table 2 materials-13-03405-t002:** Print parameters and settings.

Printing Parameters	Settings
x-y resolution	150 µm
z resolution (layer thickness)	100 µm
Laser specifications	405 nm
Printing direction	0°, 45°, 90°

**Table 3 materials-13-03405-t003:** Root mean square error (RMSE) values of trueness.

Trueness	0°	45°	90°	*p* Value
RMSE (mm)	0.129 ± 0.006 ^a^	0.086 ± 0.004 ^b^	0.109 ± 0.005 ^c^	0.001 *

Data are presented as mean ± SD. * One-way ANOVA among three groups yielded *p* < 0.05. ^abc^ Different superscript letters indicate statistically significant differences based on Tukey’s highly significant difference (HSD) test for the post hoc comparison test at *p* < 0.05.

**Table 4 materials-13-03405-t004:** RMSE values of precision.

Precision	0°	45°	90°	*p* Value
RMSE (mm)	0.072 ± 0.004 ^a^	0.050 ± 0.003 ^b^	0.069 ± 0.002 ^c^	0.001 *

Data are presented as mean ± SD. * One-way ANOVA among the three groups yielded *p* < 0.05. ^abc^ Different superscript letters indicate statistically significant difference based on Tukey’s highly significant difference (HSD) test for the post hoc comparison test at *p* < 0.05.

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
