# Peer review of "Effect of Printing Direction on the Accuracy of 3D-Printed Dentures Using Stereolithography Technology"

_materials, 2020, doi:10.3390/ma13153405_

Round 1

Reviewer 1 Report

This is a very interesting and valuable manuscript concerning the use of 3D printing technology in dental prosthetics. The work was well designed and performed, and the results were well presented. In my opinion, the authors in the introduction should also mention other types of additive manufacturing technology (AM), such as SLM: Revilla-León M et al: Int J Comput Dent. 2019; 22 (1): 55-67, Janeczek et al. RSC Adv., 2018, 8, 15533-15546; Multi-Jet Modeling: Liang et al. J Biomed Nanotechnol., 2018;14(8):1477-1485 or 3DP: Zhou et al. Mater Sci Eng C Mater Biol Appl. 2014 May 1;38:1-10. Moreover, Figures 5 and 6 should not be included in the discussion but in “results” and in “introduction” sections, respectively.

Reviewer 2 Report

Dear Authors. Thank You for the well-prepared manuscript, however, I would recommend making some changes before publication.

Line 18: scans, but not DENTURES were output as STL files

Line 19: the term “data sets” would be more precise in comparison to “data”

Line 37: are crowns and bridges devices???? You should rephrase it.

Line 46: you need the reference for „7–8 hours, which is not economical”.

Lines 71-75: format the text according to the requirements.

Line 85: is the sample size n=6 sufficient for comparisons? How it was calculated. It should be explained, as the sample size is relatively small. You should demonstrate that it is sufficient for statistical comparison.

Line 112: the same “output as STL files”; probably not dentures, find the more precise expression-word-phrase. You scan dentures, and scanning data was saved/output as STL files.

Line 113-115: the sentence should be rewritten; now it is difficult to understand what are you trying to say.

Line 206: you DO NOT need mention tables and figures in the discussion as it is the matter of the results; just mention findings and compare them with previously published data.

Don’t use the figures/tables in the discussion – it can be used in the results.

Reviewer 3 Report

The manuscript needs to update references

Reviewer 4 Report

The manuscript describes a highly systematic, well-designed and rigorous analysis of the influence of print angle on trueness and precision for dentures. It is a clearly valuable investigation within the  specific field of dental prosthetic devices, but has broader interest to the field of additive manufacturing.

A few minor specific points to be addressed:

Introduction

It would perhaps have been useful to include a schematic of vertical, horizontal and 45 degree printing directions in the introduction.

Can some inclusion of the allowable tolerances within dentures be included and the effect of ill fitting dentures mentioned. It's useful to have a point of reference as to how significant the deviation in trueness and precision actually is.

Discussion

I wouldn't describe this as an in vitro experiment, as this normally refers to a biological study (with cells). Are you instead suggesting that it is not a commercial system nor printing on the sort of scale that might be expected for denture production?

In the same way commerical production is not in vivo.

Some discussion on the following could also be included.

What are the implications of trueness and precision – do any of the printing directions result in dentures out of scope, that would be unable to be used?

How do the different printing directions affect the ease of printing, is there any reason to not implement the 45 degree printing angle? How does the support structure differ are there any challenges associated with this?
